# Epigenetic Mechanisms and Posttranslational Modifications in Systemic Lupus Erythematosus

**DOI:** 10.3390/ijms20225679

**Published:** 2019-11-13

**Authors:** Elkin Navarro Quiroz, Valeria Chavez-Estrada, Karime Macias-Ochoa, María Fernanda Ayala-Navarro, Aniyensy Sarai Flores-Aguilar, Francisco Morales-Navarrete, Fernando de la Cruz Lopez, Lorena Gomez Escorcia, Carlos G. Musso, Gustavo Aroca Martinez, Henry Gonzales Torres, Anderson Diaz Perez, Andres Cadena Bonfanti, Joany Sarmiento Gutierrez, Jainy Meza, Esperanza Diaz Arroyo, Yesit Bello Lemus, Mostapha Ahmad, Roberto Navarro Quiroz

**Affiliations:** 1Faculty of Basic and Biomedical Sciences, Universidad Simon Bolivar, Barranquilla 080001, Colombia; fdelacruzlopez10@gmail.com (F.d.l.C.L.); lorenagomez1212@gmail.com (L.G.E.); garoca1@unisimonbolivar.edu.co (G.A.M.); hgonzalez11@unisimonbolivar.edu.co (H.G.T.); adiaz72@unisimonbolivar.edu.co (A.D.P.); cadenabonfanti@gmail.com (A.C.B.); jsarmiento18@unisimonbolivar.edu.co (J.S.G.); jainymeza@gmail.com (J.M.); belloyesit0926@gmail.com (Y.B.L.); mostapha.ahmad@unisimonbolivar.edu.co (M.A.); 2School of Medicine, Universidad de Guadalajara, Jalisco 44100, Mexico; chavezestradavaleria@gmail.com (V.C.-E.); kari.macias11@gmail.com (K.M.-O.); mariafernandaayalanavarro@gmail.com (M.F.A.-N.); 3School of Medicine, Universidad Veracruzana, Veracruz 41211, Mexico; aniyensy_16@hotmail.com; 4School of Medicine, Universidad Autónoma de Guerrero, Guerrero 39350, Mexico; francisco.octu97@gmail.com; 5Department of Nephrology, Hospital Italiano de Buenos Aires, Buenos Aires B1675, Argentina; carlos.musso@hospitalitaliano.org.ar; 6Department of Nephrology, Clinica de la Costa, Barranquilla 080001, Colombia; 7Faculty of Health Sciences, Corporacion Universitaria Rafael Nuñez, Cartagena de Indias 130001, Colombia; 8Department of Nephrology, Universidad de la Costa, Barranquilla 080001, Colombia; ediaz5@cuc.edu.co; 9CMCC—Centro de Matemática, Computação e Cognição, Laboratório do Biología Computacional e Bioinformática—LBCB, Universidade Federal do ABC, Sao Paulo 01023, Brazil; Roberto.navarro@ufabc.edu.br

**Keywords:** posttranslational modifications, epigenetic mechanisms, systemic lupus erythematosus, ubiquitination, SUMOylation, glycosylation, hydroxylation, phosphorylation, sulfation, acetylation

## Abstract

The complex physiology of eukaryotic cells is regulated through numerous mechanisms, including epigenetic changes and posttranslational modifications. The wide-ranging diversity of these mechanisms constitutes a way of dynamic regulation of the functionality of proteins, their activity, and their subcellular localization as well as modulation of the differential expression of genes in response to external and internal stimuli that allow an organism to respond or adapt to accordingly. However, alterations in these mechanisms have been evidenced in several autoimmune diseases, including systemic lupus erythematosus (SLE). The present review aims to provide an approach to the current knowledge of the implications of these mechanisms in SLE pathophysiology.

## 1. Introduction

Vertebrates not only can regulate their gene expression at different levels (transcriptional and posttranscriptional) through epigenetic regulation mechanisms that involve interdependent processes such as DNA methylation, RNA-mediated epigenetic regulation, and posttranslational modifications in histones that, moreover, include the histone variant H2A.Z in Nucleosome-depleted regions (NDR) but also can modulate critical protein functions that are already expressed by increasing the response capacity of an organism genome to the different changes encountered throughout its life through posttranslational modifications. This opens up a complex universe of understanding on how a cell works and how these mechanisms alter physiologically altered states such as autoimmunity.

Systemic lupus erythematosus (SLE) is a multifactorial autoimmune disease with a wide range of clinical manifestations and severity; the mechanisms responsible for loss of immune tolerance remain to be elucidated [1], in which both epigenetic mechanisms and Post-Translational Modifications (PTMs) are increasingly recognized as important factors in SLE pathogenesis. For example, only 20% of the concordance for SLE has been found in homozygous twins, demonstrating the role played by the environment in the modulation of epigenetic mechanisms and PTMs in the appearance of SLE [2]. Another indirect evidence of epigenetics’ role is that SLE is predominantly found in women and that it can be accelerated by environmental triggers, such as infection, UV, and medications, and/or internal factors, such as hormones and stress. In addition, certain types of drugs, such as 5-azacitidine (5-azaC) and Pca, reportedly can induce SLE, causing epigenetic modifications [3]. Chromatin methylation plays a definite role in SLE [4]. The analysis of the complete genome has revealed the hyperacetylation of histone H4 in monocytes and the visualization of H3K4me3 and H3K27me3 in specific SLE genes [5]. Moreover, the activity of lupus disease and its progression have been negatively associated with methylation patterns [6,7]. The involvement of histone acetylation in SLE pathogenesis is supported by a global hypoacetylation of Cluster of Differentiation 4 (CD4)+ T histones from patients with SLE [5]. Some research shows that epigenetic mechanisms affect the expression of genes that regulate the function of cells within the immune system [8,9]. A recent study analyzed a large-scale DNA methylation between CD4+ T-cells controlling SLE and virgin cells, identifying 47 differentially methylated genes, including *BST2*, *IFI44L*, and *STAT1* [10]. Much of what is known about epigenetic regulation is the product of cancer biology research. Additionally, a key feature in the initiation of SLE is the PTMs of antigens, which result in the recognition of host proteins as “non-self” or “dangerous”, and, thus, in the initiation of an adaptive immune response, and autoreactivity to histones is a pervasive feature of SLE [11,12].

Given the complexity of the topic, this review aims to illustrate and define in a simple way the changes of epigenetics, posttranslational mechanisms, and their relationship with the susceptibility and pathogenesis of SLE.

## 2. Epigenetics and Posttranslational Mechanisms and their Relationship with Systemic Lupus Erythematosus 

### 2.1. Ubiquitination

Ubiquitin is a small regulatory and highly conserved protein that exists in all eukaryotic cells [13]. Ubiquitination is the process by which cells discriminate proteins that will be degraded [14]. Molecularly, the ubiquitin system is composed of three enzymes, namely E1 (activation), E2 (conjugation), and E3 (ligase). The first step of ubiquitination involves the formation of thioester bond with the glycine residue of the C-terminal of ubiquitin and the hydrogen sulfide group of E2 cysteine at its active center. Second, ubiquitin is converted from an E1 enzyme into an E2 conjugation enzyme. Finally, E2-Ubiquitin binds to an E3 ligase, catalyzing the formation of an isopeptide bond between the glycine of the C-terminal of ubiquitin and the lysine of the specific substrate [15].

The E3 enzymes recognize the specific protein that will be utilized during ubiquitination. Polyubiquitin chains formed by various linkages are characterized by different structural and functional information. The location and character of protein processing are determined by the diverse lysine residues that link ubiquitin chains. Specifically, K48 chains direct their linked protein substrates to degradation by proteasome 26S [16]. Polyubiquitin chains linked through K63 or K6 perform diverse functions such as DNA damage repair, endocytosis, cellular signaling, intracellular trafficking, and ribosomal biogenesis [17]. Polyubiquitin chains that are linked by K63 and K48 participate in innate immune responses through the activation of pattern recognition receptor, resulting in the activation of nuclear factor kappa-B (NF-κB) and the induction of cytokines such as tumor necrosis factor (TNF) and interleukin-1 (IL-1) [13]. Some of the cytokines are well known for their proinflammatory effects when expressed, thereby triggering, contributing, or aggravating the chronic inflammatory status of SLE. This phenomenon causes the clinical manifestation and progression of the disease in different organs and tissues, including the kidneys, heart, lungs, brain, blood, joints, and skin.

Normally, the addition of ubiquitin molecules affects the capacity of antigen-presenting cells for the antigen processing and it improves immunological tolerance by modifying the diverse signaling pathways, thereby decreasing the activation of T-cells and promoting anergy. Decreased E3 ligase expression correlates with immunity loss. The dysfunction of E3 ligases can indistinctly produce lymphocytes to activate indiscriminately and to diminish their tolerance to self-antigens [13].

Casitas B-lineage lymphoma (Cbl) comprises a family of proteins that bind to other molecules to cause its ubiquitination and degradation. In mammals, Cbl is coded by three genes, namely *c-cbl*, *cbl-b*, and *cbl-3*. In T-cells, the proteins c-cbl and cbl-b are in charge of the signaling control generated by T-cell receptor (TCR) activation by means of the ubiquitination of active receptors and tyrosine kinase-associated receptors [14].

The CD28 molecule is one of the most important co-stimulatory receptors described in T-cells essential for the complete activation of these cells. Although the activation of T-cells can occur with a signal from their TCR, the binding with CD28 is necessary in most of the responses to an antigenic peptide. The binding of phosphatidylinositol 3 kinase (PI3K) to the phosphorylated motif of CD28 triggers the production of phosphatidylinositol biphosphate and phosphatidylinositol triphosphate, which then bind to the homologous domains in proteins, such as phosphoinositide-dependent protein kinase 1 (PDK-1), which, in turn, activates protein kinase B (PKB). Both PDK-1 and PKB can phosphorylate other proteins and regulate multiple pathways linked to protein synthesis, cell metabolism, and survival. Hence, the coactivation pathways CD28, PI3K, and PKB provide signals for an increased cellular metabolism and generate pro-survival signals that prevent T-cell apoptosis [18].

C-cbl also interacts with SH2 domain of the p85 subunit of the PI3K enzyme, thereby negatively regulating the PI3K signal of the co-stimulator of T lymphocyte signaling [19]. The role of c-cbl in patients with SLE is still unclear. Possibly, the c-cbl protein in these patients is expressed downward, thereby not fulfilling its function as a negative regulator of PI3K. In this way, the pathways associated with phosphatidylinositol would be allowed to advance, thereby indirectly promoting signaling pathways that culminate in the survival of T-cells, especially self-reactive strains.

The process of neutrophil extracellular traps (NETs) has been associated with the pathogenesis of SLE, considering that NETs are extracellular fibers composed mainly by nucleic acids and can cause exposure of extracellular DNA. Components present in NET are susceptible to posttranslational modifications (PTMs), such as ubiquitination (Figure 1). This process, as seen before, can produce loss of tolerance to self-antigens, regulations in the immune system, survival of signals to T-cells, and a signaling cascade that promotes numerous intracellular changes when binding to its receptor CXCR4 [15].

Some investigations have recently discovered polyubiquitinated proteins in NETs that differ between patients with SLE and the healthy individuals. Apparently, patients with SLE had the lowest expression of ubiquitinated proteins, indicating that its higher expression in healthy patients could signify that NETs constitute a regulatory pathway to reduce the proinflammatory environment that lead to NET formation (NETosis). As described previously, K63 ubiquitination is associated with DNA repair, signaling through NF-κB, endosomal traffic regulation, and immune modulation, among many others that could conclude in the resistance of different cells to oxidative stress [20,21]. The findings of less K63 polyubiquitination in patients with SLE could lead to a higher oxidative stress in different systems and could contribute to endothelial damage and the proatherogenic phenotype and chronic inflammation in these patients [15].

Myeloperoxidase (MPO) is one of the most abundant NET proteins in which its enzyme, which is present in neutrophil phagosomes, plays an important role in autoimmunity and inflammation. NETs containing ubiquitinated MPO could enhance the production of more autoantibodies against the native and ubiquitinated enzymes in patients with SLE compared with those in healthy individuals [15]. These findings propose that the specific PTM contributes to making native enzymes more immunogenic and, consequently, greatly predisposing the T lymphocytes to be reactive to their own antigens, causing destruction in the different organs and systems. Such destruction is characteristic of patients with SLE.

Moreover, SLE macrophages internalize NETs when stimulated with lipopolysaccharide, resulting in a higher production of cytokines, especially IL-10 and TNF-a, by phagocytes [15].

Overall, abnormalities in PTMs, especially in ubiquitination, could play an important role in SLE pathogenesis and the development of inflammatory environments that translates in the different clinical manifestations in patients with this condition. SLE is not confined to a specific organ or system; it is controllable but potentially deadly. More studies are essential to understand the loss of tolerance, to clarify the triggering factor for the expression of this disease, and to be able to guide new therapies that aim to cure patients with such conditions.

### 2.2. SUMOylation

SUMOylation is a posttranscriptional modification that involves the attachment of a Small Ubiquitin-Like Modifier ( SUMO )molecule to a lysine residue in the desired protein. Though similar to what occurs in ubiquitination, both pathways involve different enzymes and produce different cellular effects. SUMOylation is involved in various biological processes, such as nuclear organization, DNA replication and repair, transcription, cellular reproduction, and signaling pathways [22,23].

The process of adding a SUMO moiety to a lysine residue involves E1, E2, and E3 enzymes. First, free SUMO is transferred to an E2 ligase enzyme or best known as UBC9, which has a catalytic function and conjugates SUMO to an acceptor lysine. Once this step is complete, the substrate conformations change, leading to diverse functional consequences, where we can find protein–protein interactions, transcription, genomic stability, and intracellular trafficking, among many others. Of note, this PTM can be reversed by SUMO-specific proteases (SENPs) [22,23,24].

Treg lymphocytes are a subpopulation of TCD4+ lymphocytes that actively suppress the pathological and physiological immune responses and, consequently, contribute to the maintenance of immunological self-tolerance and to immune homeostasis [25]. Foxp3 is a specific transcription factor necessary to differentiation, survival, and suppressor function in all Treg cells. In both mouse and human models, Foxp3 mutation or deficiency causes an early-onset and fatal autoimmunity. Once Treg cells are transported from the thymus to the periphery, they must reproduce and differentiate into effector Treg cells to prevent autoimmunity or exaggerated immune response [22].

The involvement of SUMOylation in Treg cell regulation and the implications of its deficiency are still poorly studied and understood. PIAS1, which is a SUMO E3 ligase, inhibits Treg cell differentiation by maintaining a repressive chromatin state at Foxp3 promoter [22]. Experiments on mice with SUMO-conjugating enzyme UBC9 (UBC9)-deficient Treg cells present defects in the homeostatic proliferation, alterations in the activation, and reduction of their suppressive capacity. The expression of the TCR-dependent gene in Treg cells is regulated for the SUMOylation mediated by UBC9 with Interferon Regulatory Factor 4 (IRF4) stability [22]. IRF4 is a cooperative transcription factor of Foxp3 and is necessary for Treg cells to control the response of the Th2 lymphocytes, which is a cell line responsible for the humoral immune response and the elimination of extracellular pathogens and for the differentiation and effector functions of the same Treg cells. The expression of this transcription factor was reduced in mice with absent or deficient UBC9 [22,26].

Therefore, SUMOylation can be a gene repression pathway that prevents or limits the differentiation of lymphocytes T CD4+ to Treg cells. These modifications would decrease the number of effector Treg strains and, subsequently, their repressor function in the immune response. In this way, a tolerance imbalance could exist toward self-antigens by other T-cell strains and the immune system would develop an exaggerated immune response and creation of autoantibodies, which are characteristic of patients with SLE and other autoimmune disorders. Further studies are necessary to clarify the relationship between Treg cells and SLE origin.

The acquisition of SLE is multifactorial. However, the strongest risk factor is being a female; this disease develops in women nine times more frequently than men. The female sex steroid estradiol may contribute to the gender specificity in SLE, although the exact physiopathology is still poorly understood. Normally, estradiol diffuses into target cells and binds to estrogen receptors (ERs) in the nucleus. ERα and ERβ serve as members of a transcription factor family regulated by a nuclear receptor ligand. When activated, these receptors interact in specific DNA sites and alter the transcription rate [23].

Both ERα and ERβ are expressed in human T-cells and are subjected to PTMs, such as SUMOylation. SLE is characterized by the production of antibodies that recognize self-antigens, eventually leading to the destruction of various organs and systems. TCD4+ cells, T helper 2 (Th2), and follicular helper T-cells in greater proportion are required for the generation of germinal centers that culminate in the selection of high-affinity and memory B cells. T-cells in patients with SLE have transduction pathways that are altered by estradiol. When bound to its receptors, estradiol could activate and repress genes in the same transduction pathway. Two of the most highlighted molecules were calcineurin and CD154, of which the expression levels were increased in SLE T-cell samples but not in healthy individuals. High expression of these genes in SLE T-cells is expected to enhance calcium–calcineurin, Nuclear Factor of Activated T-cells (NFAT) signaling, which is a signaling pathway that transmits signals from a wide variety of receptors to the nucleus and that is involved in the immune response. This signaling pathway finally concludes in an exaggerated help for B cells and hypersecretion of autoantibodies [23,27].

More information is needed regarding the insight into whether ER SUMOylation changes the posterior signaling pathways when exposed to estradiol and how this phenomenon would have repercussions in the transcription of genes important to Th (T helper) cell differentiation and in the exaggerated help to B lymphocytes that will lead to an overproduction of autoantibodies. This matter could be an interesting start point for further investigations.

The abovementioned mechanisms affected by SUMOylation should have a meeting point for lymphocyte differentiation to confirm whether SUMO molecules indirectly block the differentiation of the T-cell strains to Treg cells or SUMOylation causes changes in the action of ERs, leading to an overproduction of antibodies against self-antigens. However, the true pathogenesis of SLE and the molecular mechanisms that can detonate, maintain, or aggravate its development remain unclear. The PTMs have an important participation in the genesis of autoimmunity, specifically SLE. They would be an important field of research that should be further elucidated. In this way, we can identify target molecules to direct treatments that aim to improve patient’s health, quality, and life expectancy.

### 2.3. Glycosylation

Glycosylation is one of the most common PTMs that contribute to many biological processes [20]. This enzymatic PTM involves the covalent addition of sugar moieties to the protein [19]. Two known types of protein glycosylation are as follows: *O*-glycosylation, which consists of glycans linked to oxygen atoms of the amino acid serine (Ser) or threonine (Thr), and *N*-glycosylation, which is linked to the nitrogen atom of the amino acid asparagine (Asn) [28].

Carbohydrate structures also play a key role in the antigenicity of numerous clinically important antigens in adhesion and homing events during inflammatory processes; they comprise robust xenotransplantation antigens and may provide targets for tumor immunotherapy. Additionally, glycosylation is altered in many autoimmune diseases, such as SLE [29].

Immunoglobulin G (IgG) is the most abundant class in the human antibody pool and has important functions in the adaptive immune response. The antibody is separated into two functional units, namely the antigen-binding fragment (Fab) and the crystallizable fragment (Fc). IgG Fab recognizes and binds complementary structures of the antibody’s antigen, whereas sequences of IgG Fc are more conserved and activate immunological cells or molecules [30].

Fc regulates the functions of immunoglobulins on its immunological effects. The most prominent Fc encompasses the activation of Fc receptors and the classical pathway of the complement system, resulting in proinflammatory reactions [31]. Fc shows one conserved glycosylation site at asparagine 297 (Asn-297). Depending on the presence and composition of Fc, antigen binding and effector functions such as phagocytosis, complement activation, and inflammatory processes are induced with varying effectiveness [32]. The changes in this conserved site for glycosylation include the interactions with its receptors, leading to flaws in the immunocomplex (IC) clearance mechanisms, indicating its response to the mesangial deposit [21].

An increase in agalactosyl glycoforms of IgG is associated with rheumatoid arthritis (RA) and SLE. The different IgG glycoforms are produced by the diminished activity of galactosyltransferase in lymphocyte B [33]. Circulating IgG complexes are composed of a plethora of potentially glycosylated constituents and are prone to partial redirection of the IC from Fc receptors to the surface lectins of effector cells, such as monocytes, macrophages, or dendritic cells, in immunity mechanisms [34].

In SLE, antigens derived from apoptotic cells stimulate immune responses that lead to IC formation. Depositing ICs in the tissues causes clinical manifestations, such as nephritis, pleuritis, vasculitis, and skin disorders. A deglycosylated Fc considerably decreases its affinity to activate FcγRs, thereby stimulating an inflammatory immune response [35].

Apparently, inappropriate glycosylation of IgA causes conformational changes, leading to an increased IC formation. Thus, such aberrantly glycosylated IgA antibodies not only can perform their normal functions but also can become drivers of IC diseases. IgG can be abnormally glycosylated in patients with SLE [36], similar to α2-macroglobulin. Moreover, changes in the carbohydrates of IgA1 can affect the interactions between receptors and extracellular proteins, resulting in IgA IC formation and mesangial deposition and subsequently causing proliferation of mesangial cells and induction of inflammatory responses [37].

### 2.4. Hydroxylation

Protein hydroxylation is a reversible PTM in which a hydroxyl group (–OH) is introduced into amino acid residues, mainly proline and lysine. The reaction is catalyzed by hydroxylases, which are members of a large class of enzymes known as 2-oxoglutarate-dependent dioxygenases, and it plays several critical roles in biological systems [38].

Proline hydroxylation is important for the activation of antioxidant defense against hypoxia through the hypoxia-inducible factor (HIF) [38]. HIFs are transcription factors that coordinate cellular responses to low oxygen levels, aiming at increased oxygen delivery and reduced oxygen consumption [39].

In normoxic conditions, prolyl hydroxylase domain proteins can lead to the hydroxylation of HIF-α and the degradation of the proteasome. Under hypoxic conditions, HIF-1α is stabilized and the HIF heterodimer with HIF-1β is formed. Collectively, these conditions endorse the transcription of hypoxia-response genes, with functional consequences such as angiogenesis, metabolic adaptation, metastasis, apoptosis, and other physiological processes [38].

Estrogens can promote autoimmunity, leading to an indirect increase of inflammation, whereas androgens generally suppress autoimmunity. Estrogens increase the production of autoantibodies. In SLE, estrogens are metabolized differently. Considering the abnormality in a chemical pathway called 16 alpha-hydroxylation, patients with SLE have excess levels of 16 alpha-hydroxyestrone [31].

Clearly, estrone is preferentially hydroxylated at the C-16 position in patients with SLE, resulting in the accumulation of substances that have a high degree of estrogenic activity [40]. The compound 16α-hydroxyestrone is uniquely estrogenic. Patients with Klinefelter syndrome and SLE also have an accumulation of the 16α-hydroxylated metabolites. Thus, the estrogenic products of estrone hydroxylation in patients with SLE are increased significantly [40].

Some of the characteristic symptoms of SLE are associated with autoantibody production against a myriad of nuclear antigens. Oxygen-derived free radicals are involved in the pathology of SLE, particularly in processes leading to the formation of pathological anti-DNA antibodies [39].

Reactive oxygen species (ROS) such as O_2_^−^, H_2_O_2_, and ·OH are produced during normal cellular metabolism but can be augmented after cellular exposure to some factors, including UV radiation, carcinogens, cellular photosensitizers, and certain metal catalysts [41]. Excess ROS has been implicated in the etiology of many human diseases, including SLE, because they may cause damage to DNA and other macromolecules; autoantibodies to a self-antigen are also produced [39]. A hydroxyl radical (OH) can interact with chromatin, resulting in a wide variety of sugar products as well as DNA-protein crosslinks; furthermore, single- and double-stranded breaks show the mutagenic and carcinogenic potential of these radicals [41]. The serum of patients with SLE has been examined on the binding of SLE autoantibodies with native and ROS-modified chromatin. A chromatin modified by oxygen free radicals exhibits a preferentially high binding with SLE. Therefore, the chromatin after being modified by OH, superoxide anion radical, and singlet oxygen presents unique antigenic determinants for the production of SLE autoantibodies [39].

Guanine is highly susceptible to modification by OH, leading to the formation of 8-oxodeoxyguanosine [39]. This 8-oxodeoxyguanosine is found in the DNA of patients with SLE and is considered as a marker for oxidative DNA damage. Lymphocytes isolated from patients suffering from RA and SLE contain increased levels of 8-oxo-dG in DNA, suggesting that ROS damage may lead to the formation of modified guanine moieties in DNA and RNA. Such formation influences their immunogenicity, consequently inducing the production of anti-DNA and anti-RNA antibodies in patients with SLE and primary Sjogren’s Syndrome (pSS).

### 2.5. Phosphorylation

Phosphorylation is an enzymatic process by which a phosphate group is added to a small molecule or protein through an ester bond. This addition modifies the hydrophobic apolar protein into a polar hydrophilic, allowing the protein to change conformation when it interacts with other molecules. For example, a phosphorylated amino acid can bind molecules that are able to interact with other proteins, leading to the assembly and separation of protein complexes. Many of the cell phosphate esters are phosphoproteins that are formed by means of a catalytic enzyme and adenosine triphosphate (ATP), which is a phosphate anhydride that acts as a donor of a phosphate group [42].

Protein phosphorylation is important in regulating signal transduction systems, given that it affects the activity of its target, such as in turning an enzyme “on” or “off”. Furthermore, it is an extremely important regulatory mechanism in most cellular processes, such as cell division and growth, protein synthesis, signal transduction, development, and aging [42].

Many enzymes and receptors are activated and deactivated by events such as phosphorylation or dephosphorylation due to specific kinases which catalyze the aggregation of phosphate groups and phosphatases which catalyze the elimination of phosphate groups; thus, phosphorylation is considered a dynamic and reversible process. The human genome consists of approximately 568 protein kinases and 156 protein phosphatases, which are responsible for controlling phosphorylation events, thereby playing an important role in the control of biological processes, such as proliferation, differentiation, and apoptosis [42].

Metabolic regulation is particularly important in autoimmune diseases. Phosphorylation can have important consequences for the compaction of chromatin through charge changes. The role of this modification has not been rigorously demonstrated in vitro, but demonstrations of its role in mitosis, apoptosis, and gametogenesis are indicative of such a role. Phosphorylation of histones may not be as analyzed as methylation because different signaling pathways must be activated to observe the modifications.

In addition to the charge change, the chromatin compaction by phosphorylation can be modulated by regulating the binding or separation of other proteins, such as transcription factors. The phosphorylation of Ser 10 in histone H3 (H3S10P) is recognized by a family of proteins present in all investigated eukaryotic organisms and has diverse functions, such as signal regulation in different processes where transcriptional activation can be found. H3S10 phosphorylation is related to the following two roles dependent on the cell cycle: regulator of transcription during interphase together with the acetylation of different residues and condensation of chromatin during mitotic and meiotic cell divisions. If these processes are deregulated by modification, the stage of transcription in protein synthesis is altered; therefore, cellular programming is modified [43,44].

Normally, resting lymphocytes generate energy through oxidative phosphorylation (OXPHOS) and fatty acid decomposition. After activation, they rapidly change to aerobic glycolysis and low tricarboxylic acid (TCA) flow. In contrast to healthy lymphocytes, SLE T-cells secure ATP production through OXPHOS rather than upregulation of aerobic glycolysis [45].

Splenocytes from SLE mice increase glucose oxidation by 40% due to the increased activity of TCA cycle activity; thus, the glycolytic activity in chronically stimulated human T-cells can be significantly lower than that in acutely activated cells. Underlying mechanisms are unknown, but the reduction of CD28 (co-stimulatory receptor necessary for the complete activation of T lymphocytes and the production of interleukin 2) expression is related to a decrease in the activity of aerobic glycolysis. SLE T-cells have a high mitochondrial membrane potential, produce more ROS, and have a reduced intracellular glutathione, possibly caused by the acceleration of the TCA cycle; as a result, ROS are excessively generated due to the filtration of the electron transport chain. Thus, SLE is a disease associated with an increase in oxidative stress and an excessive oxidative capacity that has been related to the underlying immune dysfunction, the production of autoantibodies, and the cardiovascular complications of the disease. Dysfunctional mitochondria have been reported as the main source of excess ROS in SLE [45].

Moreover, a significant difference exists between lipid metabolism and T-cell dysfunctions in SLE. In TCD4+ cells of patients with SLE, glycosphingolipids are associated with significantly elevated lipid rafts compared with those of healthy individuals. In the same T-cells, the expression of the liver X receptor is increased. The X receptor is a family member of transcription factor receptors that function as important regulators of the homeostasis of fatty acids and cholesterol. Alteration of glycosphingolipids and cholesterol homeostasis in lipid rafts can lead to abnormal signaling of the TCR, probably by increasing the critical signaling mediators such as the Protein Tyrosine Leukocyte Specific Tyrosine Kinase (LCK) and CD45 [45]. LCK plays a key role in the activation and differentiation of T lymphocytes. It is also associated with various cell surface receptors and is critical for the early propagation and modulation of the TCR signaling. Meanwhile, CD45 inhibits the activity of LCK kinase [46].

Therefore, an inhibition of glycosphingolipid metabolism can normalize TCD4+ cell signaling and decrease the production of antibodies against double-stranded DNA by autologous B cells. Probably, lipid biosynthesis is closely related to membrane function, but the molecular mechanisms that drive the metabolic dysfunction of lipids in T-cells in SLE remain unclear [45].

Owing to the increased production of the glycosphingolipids lactosylceramide, globotriaosylceramide, and monosialotetrahexosylganglioside, SLE T-cells change the membrane raft formation and fail to phosphorylate pERK (a protein kinase R-like endoplasmic reticulum kinase) yet hyperproliferate. Considering that pERK fails to activate, the eukaryotic initiation factor (eIF) is not phosphorylated in SER 51; hence, translation will not be inhibited and cell protection against irreversible damage caused by the accumulation of proteins deployed in the endoplasmic reticulum will be eliminated [45].

A metabolomic study comparing the sera of 20 patients with SLE with those of healthy controls was conducted using Liquid Chromatography–Mass Spectrometry (LC/MS) and Gas Chromatography–Mass Spectrometry (GC/MS) platforms. Key differences were validated using orthogonal assays and an independent cohort of 38 patients with SLE. Results showed that SLE sera showed evidence of a profound reduction of glycolysis, the Krebs cycle, the β-oxidation of lipids, and the set of available amino acids. If these changes are seen at the intracellular level, the generation of ATP is drastically reduced, conforming to previous reports that indicate that SLE T-cells are less efficient in generating ATP. This finding may be one of the molecular explanations of the chronic fatigue that patients with SLE normally suffer, although it must be formally proven [47].

### 2.6. Sulfation

Tyrosine (Tyr) sulfation sites are placed in proteins that have a key role in the development of autoimmunity and are considered one of the most common PTMs in multicellular eukaryotes. Tyrosine sulfation can regulate mononuclear cell function at various stages of an immune response. This modification is performed by two membrane-bound tyrosyl protein-sulfotransferases (TPST 1 and 2) within the trans-golgi system, thereby having access to secreted and membrane-bound proteins. Other than the upregulated expression of TPST mRNA in the liver, these enzymes are ubiquitously distributed among most of the tissues. In the absence of any known enzyme regulatory mechanisms, tyrosine sulfation is postulated to sulfate tyrosines based on the context at which they are presented. However, the physiological importance of the existence of two TPST isoenzymes in most, if not all, cells is still poorly understood.

Several tyrosine-sulfated peptides and proteins are purified from an extracellular human source, and they are isolated using methods that are independent of the sulfation state of the protein. Based on the results of a relatively small number of these studies, which indicate the absence of an efficient protein tyrosine sulfatase activity in the extracellular space, Tyr sulfation is most likely irreversible in vivo [48,49].

Tyrosine sulfation is a key modulator of protein–protein interactions that mediate inflammatory leukocyte adhesion. The recent discovery of tyrosine-sulfated chemokine receptors suggests an even broader role in the inflammatory response [50]. In leukocyte adhesion, tyrosine sulfation of the P-selectin glycoprotein ligand-1 mediates a high-affinity binding to P-selectin; in leukocyte chemotaxis, tyrosine sulfation of chemokine receptors is required for the optimal interaction with chemokine ligands [51].

A predictive algorithm is used to accelerate the detection of tyrosine sulfation sites. A total of 62 proteins are subjected to tyrosine sulfation in addition to proteins for which the site of tyrosine sulfation remains unknown. Among these proteins, nine are cell-adhesion molecules and known chemokine receptors that are involved in leukocyte trafficking.

The chemokine receptor system is necessary for leukocyte recruitment, which is a critical step to promote SLE progression. C-C motif chemokine Receptor 5 (CCR5) is the most investigated tyrosine-sulfated chemokine receptor; it is the chemokine receptor with the most sulfated residues.

Chemokine receptors belong to a large family of G protein-coupled seven-transmembrane-segment (7TMS) receptors. CCR5 carries four tyrosine residues in the amino-terminal region, all of which show some degree of sulfation in cell culture. Incubation of CCR5-expressing Fetal Canine Thymus 2 (Cf2Th) cells with sodium chlorate, which is a global inhibitor of sulfation, decreases the binding of macrophage inflammatory proteins, namely Macrophage Inflammatory Proteins (MIP-1α) and MIP-1β, which are natural chemokine ligands for CCR5. Sulfotyrosine 14 of CCR5 plays a particularly important role in binding to the chemokines MIP-1α and Regulated upon Activation, Normal T-cell Expressed, and Secreted (RANTES). RANTES is a chemokine that signals through several G protein-coupled receptors, such as CCR5, CCR3, CCR1, and US28. The main function of RANTES is to recruit leukocytes into the inflammatory sites; it also induces the proliferation and activation of certain natural killer cells to form CC-chemokine-activated killer cells [52].

In a group of Portuguese patients, upon the removal of sulfated residues, the binding affinity of CCR5 to MIP-1α and MIP-1β decreased 5 times [53], thereby demonstrating the importance of CCR5 and its ligands in SLE pathogenesis. This pathway represents a promising avenue for innovative therapeutic developments [52].

Tyrosine-sulfated proteins play important roles in many physiological and pathological processes, including hormonal regulation, hemostasis, inflammation, and infectious diseases. Sulfation of several tyrosine residues in the *N*-terminal domain of the chemokine receptor CCR5 is required for optimal binding of the chemokines RANTES, MIP-1α, and MIP-1β [51]. The specific role of tyrosine sulfation in the function of the protein is poorly known; hence, further investigation is necessary to further investigate the role of sulfation in immunity, specifically in SLE.

Another 10 sulfated proteins are affiliated with the proteins responsible for the coagulation cascade.

The tyrosine sulfation in the mu chain of immunoglobulin M was discovered in 1988. Recently, tyrosine sulfation has been discovered in the IgG gamma chain, which is important for recognizing the neutralizing capacity of the anti-gp120 antibody with specificity for the binding domain to CCR5 [54], showing that the chemokine receptors CCR5 and CXCR4 are tyrosine-sulfated. CCR5 is a chemokine receptor that serves as a co-receptor.

### 2.7. Physiological Implications of Acetylation Proteins in SLE

SLE is a chronic and multisystem inflammatory autoimmune disease in which the immune system attacks healthy cells, thereby damaging tissues of the human body. The epigenetic factors that lead to the development of SLE include miRNAs, DNA methylation, and acetylation and deacetylation of proteins [55]. Acetylation is a process wherein an acetyl group is introduced into a chemical compound. Conversely, deacetylation functions by eliminating an acetyl group [56]. The acetylation and deacetylation of amino acid residues in the tails of histone proteins are factors that influence the chromatin structure of the eukaryotic cell and modulate the transcription of genes in different ways [57]. Histones are a group of conserved proteins associated with the DNA strand; it forms nucleosomes in eukaryotic cells to maintain stability, replication, and transcription of the DNA strand as well as to alter the expression of genes. Nucleosomes are the central structures of chromatin, formed by approximately 147 bp of DNA wrapped around a histone octamer consisting of two copies of each of the central histones, namely H2A, H2B, H3, and H4 [58].

In 1964, Allfrey first reported histone acetylation and showed that lysine acetylation is highly dynamic and is regulated by the action of histone acetyltransferases (HAT) and histone deacetylases (HDAC) [59]. Histone acetylation and lysine acetylation have different functions. The acetylation of the lysine residues may lead to the loosening of the chromatin structure, increasing the accessibility of certain transcription factors to the different promoter regions of their target genes. An abnormal imbalance between lysine acetyltransferases and lysine deacetylases leads to various diseases and/or pathologies, such as autoimmunity, diabetes, neurodegenerative disorders, cardiac hypertrophy, and cancer [60]. Meanwhile, histone acetylation can function as a binding site for other proteins that function as transcriptional coactivators [56].

HATs use acetyl CoA as a cofactor and catalyze the transfer of an acetyl group to the ε-amino group of the lysine side chains; consequently, they neutralize the positive charge of lysine, weakening the interactions between histones and DNA. HAT has two main classes, namely type A and type B [59]. Type B HAT targets the cytoplasm and acetylated free histones but not those already deposited in chromatin. The type B HAT-acetylated H4 can be synthesized in K5 and K12, which are factors that can deposit histones [61]. Previously, studies explored the role of the interferon regulatory factor 1 (IRF1) in the hyperacetylation of H4 in patients with SLE, demonstrating that IRF1 overexpression is associated with a general increase in H4 acetylation. This process has been favored by an imbalance in the HAT dynamics, that is, HDAC promotes the acetylation of H4 [62]. The increase in the acetylation of lysine H4 could be due to global changes in the HAT/HDAC equilibrium or to the increase in the orientation of the HAT enzymes [53]. In a previous study, the interactions of IRF1 influenced the acetylation of H4 (H4ac) in SLE wherein they used monocytes and T-cells from controls and patients with SLE. They demonstrated that D54MG cells overexpressing IRF1 are associated with a significant increase in the acetylation of H4K5, H4K8, and H4K12 compared with the control cells of the D54MG vector. Meanwhile, IRF1 interacts directly with the chromatin-modifying enzymes, supporting a model in which the recruitment of specific target genes is partly mediated by IRF1 [53].

New modifications of acetylation of the C1 inhibitor (C1-INH) have been identified, thereby explaining the association of the levels of autoantibodies against acetylated C1-INH peptides with the risk of developing SLE [53]. C1-INH is an inhibitor of endopeptidase, which is a type of serine that controls the activation of the C1 complex [63]; in this study, the C1-acetylated protein levels were higher in patients with SLE than that in healthy controls, demonstrating the association between acetylated protein and SLE.

### 2.8. Methylation

Epigenetic modifications such as DNA methylation, histone modifications, and chromatin-remodeling complexes alter the genomic DNA structure and accessibility. Some putative master regulators of vascular biology, including chromatin structure remodelers, epigenetic DNA modifications (cytosine modifications, including DNA methylation, and histone modifications), and microRNAs, have recently been the subject of extensive research to determine their role in large-scale gene network regulation [64] 

DNA methylation in CpG dinucleotides is one of the epigenetic mechanisms involved in the regulation of gene expression in mammals. The methylation patterns are specific for each species and type of tissue. The involved machinery comprises different regulatory proteins, including DNA methyltransferases, putative demethylases, methylated CpG binding proteins, histone-modifying enzymes, and chromatin-remodeling complexes. DNA methylation is vitally important to maintain gene silencing in normal development [65].

Gene expression is regulated by many epigenetic mechanisms. Among these mechanisms, DNA methylation and histone modification are involved depending on the access of transcription factors. Some drugs are introduced to patients with SLE, and factors such as UV light, viral infections, environmental pollutants, and hormones have an effect on epigenetic regulation. This effect is possibly mediated by two basic mechanisms, namely the regulation of the transcriptional machinery (RNA polymerase II, coactivators, and co-repressors) and the regulation of enzymes such as methyl transferases and histone-modifying enzymes. Among the different promoters of hypomethylated genes that are important in the pathophysiology of the disease are IL-10, CD40 Ligand (CD40L), and CD70 [66].

The association observed between IL-10 and the disease activity is supported by the correlations seen between IL-10 and other disease (e.g., active renal disease) severity markers, such as erythrocyte sedimentation rate, C3, and C4. Considering that these laboratory markers are commonly used in addition to clinical features as indicators of disease activity, their correlation with IL-10 supports its potential as a marker of disease activity. Serum IL-10 is higher in patients with active nephritis compared with that in patients with inactive disease or controls; the serum IL-10 levels also correlated with anti-dsDNA antibody titers [67].

The ability of IL-10 to enhance B cell survival, proliferation, differentiation, and antibody production as well as to inhibit autoreactive B cell apoptosis may contribute to the elevation of anti-dsDNA titers in patients with SLE. Given that circulating ICs increase the IL-10 synthesis and that IL-10 can, in turn, facilitate the production of autoantibodies, IL-10 may act pathogenically in SLE by amplifying and perpetuating an inflammatory cycle. This hypothesis requires further in vitro and in vivo exploration [67].

Serum IL-37 is also higher in patients with SLE than in the control and is strongly associated with Asian ethnicity. However, IL-37 has no statistically significant association with disease activity. IL-37 is significantly reduced in patients with organ damage, but this association is attenuated in multivariable analysis. Hence, IL-10, not IL-37, may have potential as a predictive biomarker for the disease activity in SLE [67].

Some drugs, such as Pca and Hyd, could induce lupus because these drugs inhibit T-cell DNA methylation and T-cells from patients with active lupus have considerably decreased DNA methylation. Furthermore, demethylating CD4+ T-cells with DNA methylation inhibitors such as 5-azaC, Pca, and Hyd causes autoreactivity and injecting the autoreactive cells into syngeneic recipients causes a lupus-like disease. These observations suggest that T-cell DNA hypomethylation may be fundamental in treating drug-induced and idiopathic lupus [68].

Many drugs can also interact with nuclear proteins, such as histones. This interaction determines the formation of molecules that would express new antigenic determinants and that favor the production of antihistone antibodies. In addition, the molecular structures of certain drugs are similar to those of the purine bases, which would favor a cross-reactivity with DNA. The mechanism by which autoimmunity occurs with anti-TNFs differs from the mechanisms of the rest of the drugs that cause drug-induced lupus erythematosus [69].

Patterns of T-cell DNA methylation are established as the cells differentiate in the thymus, and they serve by suppressing the expression of genes that would be inappropriate for the function of any given T-cell subset. However, the cells may express transcription factors that would activate gene expression. Initial studies demonstrated that inhibiting the replication of methylation patterns during mitosis in cloned or polyclonal CD4+ T-cells with the DNA methyltransferase inhibitor 5-azaC alters gene expression, converting normal antigen-specific cells into autoreactive cells, which can respond to autologous or syngeneic macrophages lacking the appropriate antigen peptide in the binding site of the class II major histocompatibility complex (MHC) molecule, thereby becoming autoreactive [70].

The relationship between DNA methylation and the production of autoantibodies that are greatly important in SLE pathogenesis has also been investigated by a genome-wide DNA methylation approach in a large case–control cohort using whole-blood samples. The study showed a stronger differential methylation of CpG sites within MHC genes associated with anti-Sm and anti-RNP autoantibody production in SLE. Demethylation in interferon signature genes was also observed, consistent with previous reports. Traditional therapies used to treat SLE have no effect on the methylation status of the interferon signature genes [71].

The methylation of proteins specific to lupus not only plays a role in the activation pathway of the clonotypic receptor of T-cells but also acts on other receptors widely involved in the pathophysiology of the disease; these receptors include receptor B cell and Toll-like receptors present in antigen-presenting cells [72].

The first studies that examined the rotation rate of histone methylation found that the half-life of the methyl mark in the histones is equal to that of the protein itself. Therefore, this modification is reversible. Recently, an enzyme that catalyzes the removal of methylated histones in arginine has been identified, indicating that not all types of histone methylation are irreversible. Arginine methylation of histones is linked to gene activation; thus, the recruitment of histone arginine methyltransferases, such as CARM1 (arginine methyltransferase associated with coactivator) and PRMT1 (arginine protein N-methyl transferase), is part of the initiation pathway for the transcription of genes regulated by nuclear hormones [73].

Methylation of DNA and proteins is one of the most studied modifications. In fact, some drugs have a great influence on DNA methylation. However, additional studies are needed to investigate and discover more physiological implications that this modification has in autoimmune diseases, such as SLE.

### 2.9. Citrullination

Proteins are subjected to different PTMs. Citrullination is one of these PTMs. Citrullination consists of a modification of arginine that is catalyzed by peptidylarginine deiminase (PAD). PAD hydrolyzes the side chain of arginine to form citrulline. Citrullinated proteins are associated with autoimmune diseases that include RA and lupus and other pathologies [74]. In mammals, the PAD family consists of five members, namely PADs 1, 2, 3, 4, and 6, which exhibit tissue-specific expression patterns and varying in their subcellular location [75]. In one study, the subcellular distribution of PAD 5 in granulocytes HL-60 and peripheral blood granulocytes was evaluated. The expression of PADs labeled with green fluorescent protein in HeLa cells revealed that PAD 5 is located in the nucleus whereas PADs 1, 2, and 3 are located in the cytoplasm [76]. These isoenzymes play an important role in the homeostasis and hydration of the skin [75]. PAD 4 is the most studied at present, so two more potent PAD4 inhibitors (F and CI-amidine) have been developed. Cl-amidine also exhibits a strong inhibitory effect with PADs 1 and 3, indicating its usefulness as an inhibitor of PAD [75]. PAD 4 regulates the structure of chromatin and works through its ability to eliminate histones H2A, H3, and H4 [76]. Other studies claim that PADs 4, 1, and 3 produce equimolar amounts of citrulline and ammonia [75]. PADs are calcium-dependent enzymes; the activity of PAD is highly regulated by calcium [74]. The requirement for multiple calcium ions is consistent with the structure of the calcium PAD 4 complex, showing that this enzyme binds up to 5 calcium ions at sites distal to the active site. Additional studies (not shown) showed that calcium activates these enzymes by ≥10,000 times, similar to PAD 4 [75]. Furthermore, PAD removes the arginine residues in proteins and the citrulline residues Ca^2+^ in a dependent manner [76].

Through high-throughput analysis and activity-based protein profile, we identified several reversible PAD inhibitors (streptomycin, minocycline, and chlorotetracycline) and irreversible streptonigrin (NSC 95397) [74].

Anticyclic citrullinated peptide (CCP) antibodies are serological markers for RA. Up to 10–15% of patients with SLE are also positive. One study in the sera from patients with SLE and RA showed that anti-CCP was found in 68% of patients with RA and in 17% of patients with SLE. It was more frequent in patients with SLE with deforming/erosive arthritis (38%). A high level of anti-CCP was found in rheumatoid arthritis (RA) (26%) and SLE deforming/erosive arthritis (12%). High rates of anti-CCP/CAP antibodies were found in 91% of patients with anti-CCP-positive RA and in 50% of patients with anti-CCP-positive SLE with deforming/erosive arthritis [77].

The conversion of an arginine residue in a protein to a citrulline residue is a reaction carried out by enzymes called PADs. This process causes one of the terminal imide groups in arginine to be replaced by oxygen in citrulline. Citrullination offers a unique perspective on autoimmunity because PAD activity is stringently regulated yet autoantibodies to citrullinated proteins predictably arise [78].

The citrullination of extracellular autoantigens likely follows the release of NETs and associated PADs (Table 1). Autoantibodies to citrullinated histones arise in patients with RA, SLE, and Felty’s syndrome. The citrullination of the linker H1 may play a key role in NET release because H1 regulates the entry and exit of DNA from the nucleosome. The juxtaposition of citrullinated histones with infectious pathogens and complement and of ICs may compromise tolerance of nuclear autoantigens and promote autoimmunity [78].

Autoantibodies to citrullinated keratin and citrullinated filaggrin exhibit high specificity for RA. In general, anticitrullinated protein antibodies (ACPA) have high specificity and sensitivity for erosive SLE and RA. Therefore, depending on their specific target, ACPA can be used as a reliable diagnostic marker for arthritis and erosive SLE [78].

Elevated levels of citrullinated collagen II in the synovia of patients with autoimmune diseases suggested that PAD-mediated modification of cartilage in RA joints may directly contribute to the induction of autoantibodies. Conversely, the increased citrullination of autoantigens, such as myelin basic protein, is associated with impaired function of the affected organs [78].

## 3. Conclusions

Overall, abnormalities in PTMs, especially in ubiquitination, could play an important role in the pathogenesis of SLE and in the development of inflammatory environments that translate in different clinical manifestations. Additional studies are essential to understand the loss of tolerance, to clarify the triggering factor for the expression of this disease, and to be able to guide new therapies that aim to cure the patient.

The mechanisms affected by SUMOylation should have a meeting point for lymphocyte differentiation to confirm whether SUMO molecules indirectly block the differentiation of the T-cell strains to Treg cells or whether SUMOylation causes changes in the action of the ERs and therefore produces an overproduction of antibodies against self-antigens.

Glycosylation is fundamental in various activities or functions of proteins as well as in the immune response, which is closely related to the pathogenesis of lupus. Several serum proteins are affected by alterations in their glycosylation as well as the inappropriate glycosylation of immunoglobulins that cause conformational changes, leading to increased IC formation. In patients with lupus, AGP and CRP increase whereas transferrin decreases.

Protein hydroxylation plays several critical roles in biological systems. In the case of proline, such a mechanism is important to activate antioxidant defense against hypoxia, which can result in apoptosis.

The production of autoantibodies in SLE may be related to a modification with OH in chromatin, with the guanine as the most susceptible to hydroxylation, which can result in the formation of 8-oxodeoxyguanosine found in the DNA of patients with SLE.

The molecular mechanisms that impulse the metabolic dysfunction of lipids in T-cells of SLE have not yet been clarified. However, the decrease in the generation of ATP in SLE cells has been demonstrated.

The specific role of tyrosine sulfation in the function of protein remains poorly known. Hence, more studies are necessary to investigate the role of sulfation in immunity, specifically in SLE.

Meanwhile, citrullination is a PTM. Citrullinated proteins are associated with autoimmune diseases. One of the main associations is its association with the progress of SLE, where different isoenzymes play an important role in the homeostasis and hydration of the skin. Citrullination is the modification of arginine that is catalyzed by PAD, which consists of five members showing specific tissue expression patterns that vary in their subcellular location. In addition, PAD plays an important role in SLE.

Current knowledge on the true pathogenesis of SLE and the molecular mechanisms that can detonate, maintain, or aggravate its development remains limited. Nonetheless, we found that PTMs have an important participation in the genesis of autoimmunity, specifically SLE. PTMs can be an important field of research that should be further enlightened. SLE as an autoimmune disease has processes that are tremendously similar to those in RA. However, more studies are required to further investigate this type of disease.

## Figures and Tables

**Figure 1 ijms-20-05679-f001:**
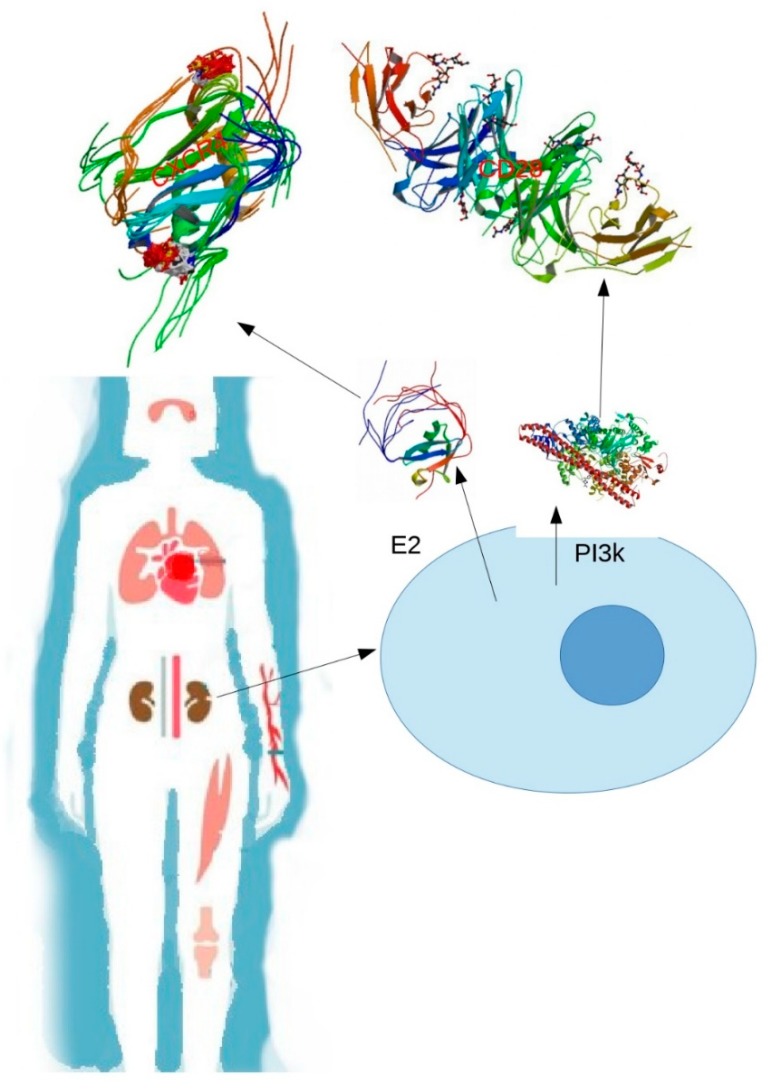
The affectation affecting several organs: for example, the kidney in the systemic lupus erythematosus can result from modifications in the proteins, leading to alterations in the tissue and loss of tolerance in these organs. The blue circle represents the cell, the E2 proteins are a ubiquinase that ubiquitin to C-X-C chemokine receptor type 4 (CXR4), and Phosphatidylinositol 3-Kinase (PI3K) is a kinase that phosphorylates Cluster of Differentiation 20 (CD20).

**Table 1 ijms-20-05679-t001:** Posttranslational regulatory mechanisms in Systemic lupus erythematosus.

Types of Modification	Protein Names	Protein Targeting	References
Ubiquitination	Enzyme 3 (E3)	K63 and K48	[20]
Ubiquitination	Cbl	zeta chain of TCR and ZAP-70	[14]
Ubiquitination	CSB6B, UBA1, and E2-25K	CXCR4	[18,60,79]
Ubiquitination	MPO	K63	[15]
SUMOylation	****SUMO E3	PIAS1	[22,23]
SUMOylation	E3	ERα and ERβ	[23,27]
Glycosylation	Glycosyltransferases	IgG	[80]
Glycosylation	Glycosyltransferases	CRP	[81]
Hydroxylation	Cytochrome P-450	16α-OHE1	[10,18,51]
Phosphorylation	PI3K	CD28	[18]
Phosphorylation	cAMP-dependent kinase	Serine 10 of histone H3	[43,82]
Phosphorylation	CD45	Tyrosine LCK kinase	[45]
Sulfation	P-selectin glycoprotein ligand-1	TPST 1 and 2	[51]
Sulfation	tyrosylprotein sulfotransferase	IgM	[54]
Sulfation	Sulfotyrosine 14	CCR5	[54]
Acetylation	HDAC	H2A, H2B, H3, and H4	[58,59]
Acetylation	Lysine Acetyltransferases	C1-INH	[63]
Methylation	CARM1	H3R17me	[73]
Methylation	PRMT1	H4R3me2	[73]
Citrullination	PAD 4	H2A, H3, and H4	[75]

Abbreviations: Enzyme 3 ligases (E3), Casitas B-lineage lymphoma (CBl), T Cell Receptor (TCR), Zeta-chain-associated protein kinase 70 (ZAP70) Chicago sky blue 6B (CSB6B), (Ubiquitin Like Modifier Activating Enzyme 1 (UBA1), Ubiquitin-conjugating enzyme (E2-25K), chemokine receptor type 4 (CXCR4), Myeloperoxidase(MPO), Protein inhibitor of activated STAT 1 (PIAS 1), Estrogen receptor alpha (ERα), Estrogen receptor beta (ERβ), Immunoglobulin G (IgG), C-reactive protein (CPR), 16α-Hydroxyestrone (16α), 2-hydroxyestrone (OHE1), Phosphatidylinositol 3-Kinase( PI3K),Cluster of differentiation antigen (CD28), Histone H3 (H3), Cluster of differentiation antigen (CD45), Lymphocyte-specific protein tyrosine kinase (LCK), Protein-tyrosine sulfotransferase 1 (TPST1), Immunoglobulin M (IgM), Chemokine receptor type 5 (CCR5), Histone Deacetylase (HDAC), Histone H2A (H2A), Histone H2B (H2B), Histone H4 (H4), C1 esterase inhibitor (C1-INH), Coactivator-associated arginine methyltransferase 1 (CARM1), methylates histone H3 at ‘Arg-17’ (H3R17me), Protein arginine N-methyltransferase 1(PRMT1), histone H4 arginine 3 (H4R3me2), Peptidylarginine deiminases (PAD 4).

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
