# Peer review of "Epigenetic Mechanisms and Posttranslational Modifications in Systemic Lupus Erythematosus"

_ijms, 2019, doi:10.3390/ijms20225679_

Round 1
Reviewer 1 Report
In the current manuscript by Navarro Quiroz et al. aim to review the current understanding of the potential implications of epigenetic epigeneticmechanisms and post-translational modifications in the pathogenesis of SLESLE.
While the language used in this review appears rather appropriate at the first glance, it does not hold when studying it more thoroughly. I would advise the authors to make use of professional linguistic support in order to support readability and clarity of their manuscript.
Partly connected to the imprecise language is the appearance of a general confusion and inaccuracy concerning fundamental concepts, definitions and examples for epigenetic mechanisms, post-translational modifications, histone modifications, protein modifications and DNA methylation.
The role of B cells and other immune cells is widely neglected in this review, both in terms of studies of epigenetic alteration as well as with regard to their implication in the pathophysiology of SLE.
In many places not the original studies but rather review articles on the topic are cited. This may be acceptable to a certain point, but if possible the original articles should be referred to.
The understandability of the review might benefit from including an additional figure or table where epigenetic mechanisms and their implications for SLE could be presented in a summarized way.
In general, the whole manuscript appears rather lengthy, repetitive and with a lack of focus. It may not be necessary to explain the molecular biology of each and every modification in great detail, but instead a more critical assessment of the potential role of these mechanisms for SLE need to be given. Many of the presented studies suffer from small sample size and inconclusive results, which is important to be discussed. Also, the authors need to be clear about whether a certain study has been carried out e.g. in a mouse model, in human cell cultures or in patients with SLE, and need to interpret the results in light of these circumstances.
Regarding the role of DNA methylation in SLE the review and it's references are not up to date and not comprehensive.
Author Response
Sincerest thanks for your response and reviewers comments on our manuscript. Following your letter regarding the manuscript Epigenetic Mechanisms and Posttranslational Modifications in Systemic Lupus Erythematosus submitted to International Journal of Molecular Sciences for publication, we are sending the rebuttal letter explaining the changes performed on the manuscript. The changes incorporate the suggestions of the reviewers. We found the comments very helpful and constructive. We have addressed all the changes recommended by the reviewers and we are confident that the new version of the manuscript is easier to understand and has a more fluent scientific discourse. We will respond to the comments point counter point:
In the current manuscript by Navarro Quiroz et al. aim to review the current understanding of the potential implications of epigenetic epigeneticmechanisms and post-translational modifications in the pathogenesis of SLE.
While the language used in this review appears rather appropriate at the first glance, it does not hold when studying it more thoroughly. I would advise the authors to make use of professional linguistic support in order to support readability and clarity of their manuscript.
Answer: Professional linguistic support was used to support the readability and clarity of your manuscript..
Partly connected to the imprecise language is the appearance of a general confusion and inaccuracy concerning fundamental concepts, definitions and examples for epigenetic mechanisms, post-translational modifications, histone modifications, protein modifications and DNA methylation.
Answer: We connect and improve inaccurate language by clarifying fundamental concepts of both epigenetic mechanisms and post-translational modifications
The role of B cells and other immune cells is widely neglected in this review, both in terms of studies of epigenetic alteration as well as with regard to their implication in the pathophysiology of SLE.
Answer: The role of B cells and other immune cells is complemented, both in terms of epigenetic alteration studies and in terms of their involvement in the pathophysiology of SLE.
In many places not the original studies but rather review articles on the topic are cited. This may be acceptable to a certain point, but if possible the original articles should be referred to.
Answer: More original studies are cited
The understandability of the review might benefit from including an additional figure or table where epigenetic mechanisms and their implications for SLE could be presented in a summarized way.
Answer: Figures and tables are included
In general, the whole manuscript appears rather lengthy, repetitive and with a lack of focus. It may not be necessary to explain the molecular biology of each and every modification in great detail, but instead a more critical assessment of the potential role of these mechanisms for SLE need to be given. Many of the presented studies suffer from small sample size and inconclusive results, which is important to be discussed. Also, the authors need to be clear about whether a certain study has been carried out e.g. in a mouse model, in human cell cultures or in patients with SLE, and need to interpret the results in light of these circumstances.
Answer: Repetitive sections are eliminated and the animal model used in a large part of the cited studies is expressed.
Regarding the role of DNA methylation in SLE the review and it's references are not up to date and not comprehensive.
Answer: up date reference for DNA methyation
Reviewer 2 Report
The review manuscript entitled “Epigenetic mechanisms and post-translational modifications in systemic lupus erythematosus (SLE)” rises a very “hot” topic of epigenetic involvement in the autoimmune disease development. The presented material, though very interesting, is focused on selected aspect of epigenetics – post-translational protein modifications. This limitation is obvious, as epigenetics itself is an extremely broad and not completely uncovered issue. The Authors put a lot of effort to collect and present noteworthy data. Still the chosen subject is very complex, thus really demanding for the reader. Therefore, Authors should take into account the accuracy of the presented data along with the style of their presentation. The latter one is the main flaw of the presented manuscript. The article is difficult to read and comprehend, mainly due to the poor language and messy writing style. Particularly the first part of the manuscript is filled with language mistakes, repetitions and confusing sentences. Hence, even though absorbing, the manuscript cannot be published in the present form.
I do not have any critical comments regarding presented data. But I do strongly recommend rewriting manuscript under the guidance of some English language expert.
Below I list a few fragment of the text that caused confusion or doubts:
Abbreviations – no consistency in explanation and usage of abbreviations, some explanations are extensively used, i.e SLE, some are completely forgotten (TF, L-PGDS) PTM – nice abbreviation for post-translational modifications – why it does not appear at the beginning of the manuscript but in the fifth section? Often misused singular and plural forms Interchangeable use of past and present form of verbs – please be consistent and consequent Line 40 – “small molecules such as glycosylation..”– is glycosylation a molecule? Line 87 – “CD4 + T histones” – what is that? Line 90 – “virgin cells” – please provide definition Line 101 “Consist of” instead “conform by” Lines 102, 104 “Consist of” instead “consist in” Lines 124-125 – loss of immune…- sth is missing Lines 126-127 – a confusing sentence, I do not understand what was the Authors idea Lines 135-136 – what is “antigenic peptide”? Line 148 – “be allowing” – grammar Lines 149-150 – another confusing sentence Lines 153-156 – what was the Author thought, I do not understand this statement Line 212 – T cells are not a cell line Line 219 – T cell strain – not strain – population, subpopulation, subset Lines 231-232 – repetition about SLE – it is redundant, as details about SLE were presented in the introduction section ( same in lines 502-503) The last paragraph of 2.2 section, lines 248-256 – unnecessary summary, please move it to the conclusions (the same with summary of 2.1 subsection) The first paragraph of section 2.3 – repetition – use the abbreviation PTM Lines 261-264 – something is missing in this paragraph, as the meaning is lost Lines 285-286 – to what this sentence applies?, the same with lines 291-292 and lines 301-304 Line 360 – something is missing there – please complete this sentence and correct the subsequent Paragraph lines 365-371 – a bit confusing Line 476 – “9 out of 62 proteins…” sounds better Lines 479-483 – another confusing paragraph Lines 623-624 – citation – lost number form the reference list
Additionally, I advise to focus on the Conclusion section, as in the present form it is rather a list summarising each of the subsections not a sound recapitulation of the reviewed data.
Nevertheless, I had a great pleasure reading the submitted manuscript.
Author Response
Sincerest thanks for your response and reviewers comments on our manuscript. Following your letter regarding the manuscript Epigenetic Mechanisms and Posttranslational Modifications in Systemic Lupus Erythematosus submitted to International Journal of Molecular Sciences for publication, we are sending the rebuttal letter explaining the changes performed on the manuscript. The changes incorporate the suggestions of the reviewers. We found the comments very helpful and constructive. We have addressed all the changes recommended by the reviewers and we are confident that the new version of the manuscript is easier to understand and has a more fluent scientific discourse. We will respond to the comments:
Answer:
it is done grammatically corrected, seeking to give more clarity and coherence to the entire document
In the first paragraph, on the second occasion that post-translational modifications are mentioned, the abbreviation PTM was placed.
Line 261-264: Mention separately and define the two types of glycosylation, so that the idea is not lost. And raise the line to join with idea of the previous paragraph
Line 285-286: Place this paragraph together with the previous one, so that it is the continuous idea, and modify the way in which it is written to clarify that it refers to the changes in the site conserved for glycosylation in the FC fragment and How this relates to immune mechanisms.
Line 291-292: Change the “Being the latter” with which the sentence begins with a direct connection with the previous sentence, as well as mention that monocytes, macrophages and dendritic cells are cells that participate in immune processes.
Line 301-304: Clarify that alpha-2-macroglobulin is also aberrantly glycosylated in patients with SLE.
Line 360: Remove the space and upload line 361 since this is the continuation of what begins to be mentioned on line 360